# Critical Perspective of Animal Production Specialists on Cell-Based Meat in Brazil: From Bottleneck to Best Scenarios

**DOI:** 10.3390/ani10091678

**Published:** 2020-09-17

**Authors:** Marina S. Heidemann, Cesar A. Taconeli, Germano G. Reis, Giuliana Parisi, Carla F. M. Molento

**Affiliations:** 1Animal Welfare Laboratory, Federal University of Paraná, Rua dos Funcionários, 1540, 80035-050 Curitiba, Brazil; carlamolento@ufpr.br; 2Department of Statistics, Federal University of Paraná, Rua Cel. Francisco Heráclito dos Santos, 100, 81531-980 Curitiba, Brazil; cetaconeli@gmail.com; 3School of Business Administration, Federal University of Paraná, Av. Prefeito Lothário Meissner, 632, 80210-170 Curitiba, Brazil; glufkereis@ufpr.br; 4Department of Agriculture, Food, Environment and Forestry, University of Florence, Via delle Cascine, 5, 50144 Firenze, Italy; giuliana.parisi@unifi.it

**Keywords:** animal protection, animal welfare, animal production, cultivated meat, cultured meat, slaughter-free meat

## Abstract

**Simple Summary:**

The opinion of professionals involved in animal production is very important to the development of the emerging cell-based meat chain. This paper aims to analyse the perspective of Brazilian veterinarians and animal scientists regarding cell-based meat—women, veterinarians, vegetarians and vegans were more supportive of cell-based meat. The resistance expressed by the professionals seems related to a lack of knowledge and the association of cultivated meat with artificiality, which has a negative connotation. Therefore, higher education and motivation of veterinarians and animal scientists may mitigate the resistance and help these professionals to engage in this new chain for the benefit of the professionals themselves, society, the animals involved and the environment.

**Abstract:**

Recently, many studies regarding consumer perception of cell-based meat have been published. However, the opinion of the professionals involved in animal production also seems relevant. In particular, veterinarians and animal scientists may be important players in the new cell-based meat production, acting as proponents or barriers to this major improvement for farm animal welfare. Therefore, our aim is to analyse the knowledge and perspective of Brazilian veterinarians and animal scientists regarding cell-based meat. Veterinarians (76.8%; 209/272) and animal scientists (23.2%; 63/272) responded to an online survey. Logistic regression, latent class and logit models were used to evaluate objective answers, and the Discourse of the Collective Subject method was used to interpret open-ended answers. Specialists who were women (62.5%; 170/272), veterinarians (76.8%; 209/272), vegetarians (7.0%; 19/272) and vegans (1.1%; 3/272) were more supportive of cell-based meat. Lack of knowledge and the connection with artificiality, the most frequent spontaneous word associated with cell-based meat by all respondents, were the main negative points highlighted. Thus, it seems fundamental to offer higher education to veterinarians and animal scientists regarding cell-based meat, since engaging them with this novel technology may mitigate both the resistance and its negative consequences for the professionals, society, the animals involved and the environment.

## 1. Introduction

Currently, it is widely recognised that new technologies and systemic innovation are critical for the profound transformation the food system needs [1]. Cell-based meat is an alternative to conventional meat that does not require the husbandry and slaughtering of animals [2]. As such, there are evident benefits to farmed animals, as billions of lives may be spared the intrinsic suffering inherent to intensive industrial animal production systems and slaughter. Moreover, the increasingly questioned paradox of humane slaughter [3] may finally be completely bypassed. Indeed, the development of alternative meats may be related to a significant change in our relationship with nonhuman animals, with greater benefits than the *prima facie* effects on farm animals; this has been discussed in detail in [4].

The new cell-based technology may radically change the meat production chain that currently depends on the production of livestock on farms, their slaughter, processing, and marketing, as the new production process is based on tissue engineering, initially developed for biomedical purposes [5,6]. It begins with an animal biopsy and extraction of satellite cells, followed by cell proliferation and differentiation in a bioreactor, the final product of which is muscle tissue. Besides the already mentioned benefits to animal welfare [6], there are additional relevant gains in other areas, such as a reduction in environmental impact [7] and improvements in human health [8]. However, there are challenges yet to be overcome, such as the adjustment of the production process from small amounts for biomedicine to a meat production scale [6], alternative culture and growth media with nonanimal ingredients [5,8], price [9], flavour and appearance [10], among others. In addition, there are controversial aspects regarding social and economic implications of cell-based meat production, and the consideration of these aspects seems key to providing equality between all actors in this new chain [6]. Additionally, in terms of energy consumption, a risk of higher demand for cell-based meat production has been postulated [7].

In spite of the challenges ahead, the introduction of cell-based meat for human consumption is reportedly close to occurring [11]. Large companies of the meat industry (e.g., Tyson Foods, Cargill) have been investing in this novel technology, as well as several startups that have been created worldwide [12]. Moreover, recent forecasting reports have estimated a 30% reduction of the conventional meat in the US by 2030 [13], and by the year 2040, 35% of the global meat production may be cultivated [14]. Thus, cell-based meat is likely to bring radical changes to the meat production sector, and it seems wise to examine its implications for society in general, for animals, as well as for the global meat business.

Recent research has focused on debating consumer attitudes regarding cell-based meat. For instance, many respondents perceived the environmental, human health and animal welfare benefits of cultured meat simultaneously to a general concern in relation to the unnaturalness of cell-based meat [9,15,16,17,18,19,20,21]. However, nothing is known regarding the opinions of relevant actors in the meat chain, such as professionals and specialists who are currently involved in the production of conventional meat.

Since the introduction of cell-based meat and other alternative protein sources in the meat market is likely to occur [13], the roles and activities of those actors are most likely to be reconfigured, and new professional opportunities may arise. This is the case of veterinarians and animal scientists, who have key roles in the conventional meat production chain. For decades, both professionals have been responsible for the reproduction, growth, development and economic efficiency of farm animals, as well as the technology of meat and other animal-derived products [22,23]; more recently, animal welfare and sustainability issues have become additional responsibilities. Additionally, veterinarians have an important contribution to One Health surveillance, promoting healthy animals and controlling the processing and distribution of animal-derived products [24,25]. Then, as cell-based meat products reach the markets in the next few years, their duties may concentrate on activities such as genetics, nutrition, health, management and development of cells, as well as processing, package, marketing, control, and inspection of cultured meat products. Moreover, since the technology is just being realised, completely new roles may emerge as the new chain matures. Thus, their activities are expected to face substantial changes ahead, which will likely to be shaped by their attitudes now. However, it is not yet known how they envision these coming changes and their possible consequences. We argue that understanding their impressions is important because they may be directly involved in developing and promoting the new chain, e.g., by establishing novel cell-based meat products, by their involvement in both the creation and overseeing of regulations regarding animal products, and by their virtually omnipresent activities in the meat chain. On the other hand, they may also be a major source of resistance to change [25,26], hindering the pace of this new meat industry. In addition, the knowledge extent of this particular group, regarding cell-based technology and its implications for the meat chain, is not known; this aspect is particularly relevant given that a lack of understanding may be the major cause for innovation-specific resistance [27].

Brazil is a major producer and consumer of meat. According to the Brazilian Association of Meat Exporting Industries, the livestock sector was responsible for 8.7% of the national GDP in 2019, and around 80% of production was consumed by the domestic market, with a per capita consumption estimated at 42 kg/year [28]. However, despite this tradition in meat consumption, consumers seem increasingly open to alternative proteins. For instance, 14% of Brazilians declared themselves vegetarians in 2019, in comparison to 8% in 2012, and 63% stated their interest in decreasing meat consumption [29]. Thus, meat processing firms have diversified their product portfolio in order to offer alternative proteins. Seara, for instance, a main Brazilian producer of processed meat foods, has recently experienced success with plant-based products which have sold six times more than initially forecasted by the company [30], leading to investments towards expanding this product line. As for cell-based meat, Valente and associates [21] stated that 63.6% of Brazilian consumers would eat cultured meat, while local meat processing companies have high-quality production, distribution and marketing capabilities that enable them to successfully join ventures with cultured meat producers [31]. Consequently, the overall scenario is positive for the introduction of alternative proteins such as cell-based meat to the Brazilian market, which is likely to bring novel opportunities and changes for the activities of animal production specialists.

Thus, as the role of veterinarians and animal scientists seems relevant to the future of the cell-based meat industry and its overarching consequences. The aim of this work is to explore and compare the perception of Brazilian veterinarians and animal scientists regarding cell-based meat and to understand eventual points of resistance in order to support strategies for their mitigation. We hypothesise that these professionals are not familiar with the concept of cell-based meat and present resistance motivations to this technology.

## 2. Materials and Methods

This research was approved by the Ethics Committee on Research on Humans of the Health Sciences Sector Campus of the Federal University of Paraná, Brazil, and is registered under the number 3040865/2018.

### 2.1. Development of the Research Instrument

A Portuguese online questionnaire was developed on the Google Forms platform, with an estimated duration of 15 min per respondent, in order to evaluate the perception of veterinarians and animal scientists regarding cell-based meat. This instrument was elaborated according to a literature review on cell-based meat [6,8,32] and related questionnaires [9,15,33].

The survey was composed of a total of 47 questions: 30 multiple-choice and 17 open-ended questions. Questions on knowledge of cell-based and other meat alternatives as well as their benefits and harms, comparison between conventional and cell-based meat production systems in relation to environmental impact, human health and animal welfare, respondent views on the time for cell-based meat introduction to the Brazilian market, and the perception of cell-based meat impacts to veterinarians, animal scientists and bioprocess engineers were asked. Responses to most questions were compulsory before accessing the next one, with the exception of questions 35, 37, 39, 41, 43, 46 and 47, which were optional. Between Questions 14 and 15, there was a short explanation of the concept of cell-based meat, as follows: “*Cell-based meat is produced by cell multiplication, using cells extracted once from the live animal, later grown in the laboratory*.”

The questionnaire was refined with an online survey, followed by an interview with a committee of experts. The interview aimed to collect suggestions on both the development and the application of the questionnaire regarding possible errors, lack of clarity or ambiguities [34], as well as bias avoidance. To that end, according to Lynn (1986) [35], five specialists were interviewed: three veterinarians, one animal scientist and one bioprocessing engineer. Professionals were selected according to their experience, areas of expertise, career time and knowledge of the methodology of questionnaire construction; none of them had specific interests on cell-based meat. They received an invitation to participate in an interview including an initial application of the questionnaire and immediate discussion of the instrument, allowing for a detailed review of the method of application, duration and content of the questionnaire [36].

### 2.2. Survey Sample

The online survey was featured via social media and email in a random sampling method. Data were collected from 20 February to 1 April 2019. The total number of respondents was 297, of which 291 agreed to participate in the study. Only responses from veterinarians and animal scientists were used, eliminating 16 respondents with other backgrounds. Additionally, data from three respondents were eliminated since two did not answer all the questions and one was a student. Thus, the total respondents considered in this work was 272, most of which were veterinarians (76.8%; 209/272) and all the others were animal scientists (23.2%; 63/272). Regarding respondent gender, 170 respondents were women (62.5%) and 102 men (37.5%); 136 (65%) women and 73 (35%) men were veterinarian respondents, and 34 (54%) women and 29 (46%) men were animal scientists. In addition, 266 respondents declared their age, which varied between 22 to 71 years, with most of them concentrated in the range between 22 and 35 years of age (57.1%, 152/266). In relation to years since graduation, most respondents graduated between 0–10 (150; 55.1%) years ago, while the rest graduated 11–46 years ago (122; 44.9%). In addition, most of the respondents graduated from Southern Brazil (77.6%; 211) courses: Parana (65.1%; 177), Santa Catarina (5.1%; 14), and Rio Grande do Sul (7.4%; 20).

### 2.3. Statistical Analysis

The data related to the questions “*Do you know a way for producing meat that does not involve raising animals?” and “Do you know what cell-based, also known as lab-grown, artificial, in vitro, synthetic, clean or slaughter-free meat is?*” were submitted to logistic regression (LR) analysis. The objective was to investigate the possible association with the factors sex, age, years since graduation, profession and meat consumption. Since age and years of graduation were strongly correlated (linear correlation equals to 0.96; *p* < 0.001), we considered only years since graduation for further analyses. Initially, an LR model was fitted, including all factors under investigation; then, factors with no significant effects were successively removed. Significance was assessed by the likelihood ratio test, considering a significance level of 5%. For the variables remaining, the respective effects are presented as odds ratios and the respective confidence intervals (95%). A similar analysis was applied to the question “*Should Brazil invest in cell-based meat production?*”.

Latent class analysis (LCA) was applied to analyse two groups of questions. The first group was composed of the questions related to the benefits and harms of both conventional and cell-based production. The second group consisted of the questions regarding the environment, human health, animal welfare and efficiency of production for both conventional and cell-based meat. LCA is a multivariate statistic applied to categorised data to identify patterns in the values and groups of individuals with similar results. In the present study, the objective was to identify common patterns of responses (LCs) to the sets of related questions. Next, the respondents were grouped according to the answers to the original questions in each of the LCs obtained. Conditional to the LCs produced, the answers to the original questions were independent.

The number of LCs extracted is a key result of the analysis. Solutions starting with one to four LCs were tested and compared using the Bayesian information criterion (BIC); the solution that produced the smallest value was selected. After choosing the number of LCs, the results were plotted by means of the probabilities in each of the classes. It is noteworthy that at this stage of the analysis, as with the data extraction process, the results were classified into three categories: “positive”, “neutral” and “negative”. For example, very inefficient and inefficient were grouped into a single response. In order to identify factors associated with the LC obtained, each individual was classified into the one with the highest probability, according to his or her original answers, and the data was also submitted to LR analysis. The same method was applied regarding the knowledge of alternative sources or investment in the area. The results are presented in the form of odds ratios and the respective confidence intervals.

Finally, the answers on respondent perception regarding the probable positioning of veterinarians, animal scientists and bioprocess engineers in relation to cell-based meat were grouped into three categories: “favourable”, “neutral” and “unfavourable”. A multinomial LR, specifically the generalised logit model (LM), was used due to a variable response with the three categories. The selection of variables and the results presented were similar to those described previously, in the case of binary LR. As every respondent expressed his or her opinion for the positioning of each professional class, a random effect of respondents with normal distribution of zero mean and constant variance, corresponding to the effect of the respondent, was incorporated into the model.

All analyses were performed using the statistical environment R [37], version 3.6.0. The poLCA package [38] was used to analyse LCs and the ordinal design for the fit of generalised LMs, and the ggplot2 package was used in the construction of the plots. In addition, the arm package was used in the LR adjustment, using Bayesian estimation.

### 2.4. Analysis of Responses to Open-Ended Questions

Four open-ended questions (15, 39, 41 and 43) were analysed using the Discourse of the Collective Subject (DCS) method [39]. Question 15 was analysed individually, and questions 39, 41 and 43, which were optional and approached respondents’ hypotheses to the reasons for the probable positioning of veterinarians, animal scientists and bioprocess engineers in relation to cell-based meat, were analysed together.

DCS is a qualitative–quantitative method that uses central ideas to represent collective interpretation [39,40]. Hence, key expressions were extracted from the answers and synthesised into central ideas, which generated categories of attitudes. The frequency of a category’s appearance represents the collective opinion of the group. A single response may be present in more than one category; thus, the total frequency of category is higher than total responses.

For question 15, every word with a complete meaning as a central idea was used, such as “artificial”, “health”, and “colour”. When answers were presented as sentences, they were synthesised into central ideas. For example, “possibility of greater market acceptance” was transformed to “possibility, market, acceptance”. Adjectivised nouns (e.g., “animal welfare”) were maintained to guarantee the full meaning intended by respondents. Lastly, the website Word It Out (https://worditout.com) was used to present the words with sizes proportional to their frequency in the dataset. Only words that appeared two or more times were used; words which were mentioned only once were not used to preserve clarity within the word cloud.

For the answers on respondent perception regarding the probable positioning of veterinarians, animal scientists and bioprocess engineers in relation to cell-based meat, key expressions of respondent sentences were incorporated into central ideas and measured for their frequency. For example, “Animal production is part of the attributions of these professionals” was synthesised to the central idea of “field of work”. Thus, interpretative and theoretical validities [41] were used to establish these categories in order to minimise bias.

## 3. Results

Regarding the participants’ eating habits, 91.9% consumed meat (250/272), 65.6% (164/272) declared daily meat consumption, 7.0% (19/272) were vegetarians and 1.1% (3/272) were vegans. Regarding alternatives to conventional meat production, 122 (44.8%) respondents were not aware of any other method. In relation to Question 13 “*Do you know what cell-based, also known as lab-grown, artificial, in vitro, synthetic, clean or slaughter-free meat is?*”, 74.2% (202/272) had heard of the subject.

For the LR analysis applied to the question about alternatives to traditional meat production, the only significant variable was the frequency of meat consumption (*p* = 0.021). Individuals who did not consume meat presented higher odds of responding positively than those who consumed meat daily (OR = 3.00; CI (95%) = 1.047, 8.594). However, there was no difference between nonconsumers and daily meat-eaters in relation to those with intermediate meat consumption frequency, who consumed meat but not on a daily basis. For question 13, none of the studied variables was statistically significant. For the last question evaluated with LR, “*Should Brazil invest in cell-based meat production?*”, the frequency of meat consumption (*p* < 0.001) was again a significant variable: vegetarians and vegans presented higher odds of responding yes than daily (OR = 25,920; CI (95%) = 4,623, 265,310) and casual meat-eaters (OR = 7.4320, CI (95%) = 1.2196, 79.0199).

In relation to the LCA for the two groups of questions, the solution with two LCs (Classes 1 and 2) was the best for both, based on BIC values. The first group of questions concerns the benefits and harms related to conventional and cell-based meat (Figure 1).

Latent Class 1 (79.8%; 217/272) is defined by a pattern of responses that characterises individuals who are more likely to perceive benefits of conventional meat production (proconventional meat) and tend not to see benefits in the production of cell-based meat or express lack of knowledge about the subject. In addition, they are less likely to express problems with conventional production if compared to LC 2 responses. On the other hand, LC 2 (20.2%; 55/272) is composed of a pattern of responses that describes individuals who are more likely to express benefits in terms of cell-based meat production (pro-cell-based meat), who tend to see fewer benefits in conventional meat production. In addition, there is practical unanimity in LC 2 responses regarding the existence of problems in the conventional meat production system. Individuals of the two groups showed quite heterogeneous responses to cell-based meat harms; for this variable, it was not possible to discriminate individuals from both classes. The LR results are shown in Table 1.

There were lower odds of men belonging to LC 2, which corresponds to a trend to identify more benefits than harm in cell-based meat production and the opposite for conventional meat production (OR = 0.131). There are higher chances of veterinarians belonging to LC 2 in relation to animal scientists (OR = 5.224) and lower chances of casual and daily consumers of animal meat belonging to LC 2 when compared to those who did not consume meat (Figure 2).

For the second group of questions (Figure 3), 151 (55.5%) were classified as LC 1 and 121 (44.5%) as LC 2, and it was possible to determine the characteristics of the two LC profiles. Again, LC 1 presents a pattern of responses of individuals more likely to support conventional production. This is expressed more sharply in relation to the favourable responses to the impact on human health and to the efficiency of production. The perception of the same LC is less favourable in relation to environmental impacts and animal welfare issues relative to conventional production. As for the production efficiency of cell-based meat, LC 1 respondents show less probability than LC 2 respondents of manifesting themselves positively. Additionally, they are more likely to be unaware of the effects of cell-based meat. The opposite characteristics were seen in LC 2 respondents, identified as individuals with a more favourable perception of cell-based meat production, more knowledgeable on this subject and with a more unfavourable view of the conventional production system. This latter aspect is more evident, especially when dealing with the environmental and animal welfare impacts of conventional meat production. Table 2 presents the results of the LR adjusted to the classification of the individuals produced by the LCA.

The results indicate a lower chance of men belonging to LC 2 (OR = 0.426), as well as greater chances for veterinarians in relation to animal scientists (OR = 3.291); there is also less chance for respondents with occasional consumption and for respondents with daily consumption to belong to LC 2 in relation to those who did not consume meat (Figure 4).

The LM analysis performed was on respondent perception of the probable position professionals will assume regarding the production of cell-based meat. Two different models were fitted: first, the respondent’s profession was not considered; secondly, the interaction between the respondent’s profession and the respondent’s field of work was included. The likelihood ratio test indicated no significant difference in the adjustments for these models (*p* = 0.254), resulting in no significant interaction effect. Therefore, it can be assumed that veterinarians and animal scientists shared similar perceptions as to the future expectations of the three professions in relation to cell-based meat. Figure 5 presents the probabilities fitted by the model for the respondent’s perception of the positioning of veterinarians, animal scientists and bioprocess engineers in relation to the production of cell-based meat. The results indicate that expectations regarding the positioning were different depending on the professional category. In the opinion of interviewees, bioprocess engineers tend to be more positive towards the production of cell-based meat than veterinarians (OR = 7.224), and animal scientists are more likely to be unfavourable towards it (OR = 3.012) than veterinarians. Finally, animal scientists are expected to be far more unfavourable than bioprocess engineers (OR = 197.325).

Regarding the DCS analysis, the three words declared by veterinarians and animal scientists were counted and combined into word clouds in Figure 6. The ten most frequent words for veterinarians (V) were artificial (4.5%; 27/605), technology (3.6%; 22/605), animal (3.5%; 21/605), tasteless (3.1%; 19/605), strange (3.1%; 19/605), flavour (3.1%; 19/605), expensive (3.0%; 18/605), welfare (3.0%; 18/605), laboratory (2.8%; 17/605) and future (2.3%; 14/605). For animal scientists (A), they were artificial (8.8%; 16/181), tasteless (7.2%; 13/181), strange (6.6%; 12/181), flavour (3.9%; 7/181), technology (3.3%; 6/181), expensive (3.3%; 6/181), laboratory (2.8%; 5/181), chemistry (2.8%; 5/181), cost (2.8%; 5/181) and production (1.7%; 3/181).

Finally, Table 3 presents the DCS analysis of responses regarding respondent perceptions of potential motivations for probable positioning for veterinarians, animal scientists and bioprocess engineers in relation to cell-based meat. Veterinarians, who assume the veterinary class will be unfavourable to cell-based meat, stated reasons related to field of work (25), job losses (28) and tradition (16) as main justifications. In relation to the prediction of favourable positioning, the main reasons cited were animal welfare (15) and job offers (5). Next, veterinarians previewing the unfavourable position of animal scientists mentioned field of work (35) and job losses (25). Lastly, veterinarian respondents predicted a favourable view from bioprocess engineers due to field of work (40), opportunities (22) and job offers (19). Similarly, animal scientists estimated an unfavourable position from veterinarians because of field of work (4) and job losses (3), and, regarding a favourable view, the main reason cited was animal welfare (3). Additionally, animal scientists who predicted unfavourable positioning from their own professional class cited reasons related to the field of work (12), job losses (6) and the quality of cell-based meat (4); neutral expectations were related to the emergence of a new market (5) and animal welfare (2). Finally, this group presumed a favourable positioning from bioprocess engineers for reasons related to field of work (15) and job offers (7).

## 4. Discussion

Most respondents in this study were veterinarians (76.8%; 209/272), which may be related to the higher number of veterinarians throughout Brazil as compared to animal scientists. For example, in Parana, South Brazil, there are 11,422 veterinarians registered in the Regional Council of Veterinary Medicine in comparison to 666 animal scientists (36 CRMV-PR, personal communication, 29 October 2018). Additionally, there are 35 veterinary and 9 animal science programs in the State of Parana [42]. In Brazil, both veterinary medicine and animal science graduation programs are 5-year university degrees that follow secondary school, and both programs approach traditional meat production systems as obligatory content during graduation years. The first initiative for the teaching of cell-based production systems in the country was through a course entitled “Introduction to Cellular Animal Science”, offered for the first time in August 2020, at postgraduate level, by the Veterinary Sciences Postgraduation Program at Federal University of Parana. Thus, Brazilian-graduated veterinarians and animal scientists did not receive formal education regarding cell-based meat during their university years.

The respondent invitation methodology may have resulted in more veterinarian respondents as we were welcome by official veterinary institutions and corporations, such as the National Association of Small Animal Veterinary Clinics in Parana. Other reasons, such as the level of interest in the topic of the questionnaire, may also have played a role for a higher number of veterinarian respondents. Additionally, most respondents were women (62.5%), young (22–35 years of age: 57.1%; 152/266) and recently graduated (0–10 years: 150; 55.1%). The higher number of women and young people may be related to the involvement of these groups in animal welfare and their higher willingness to try alternative proteins [43,44]. Additionally, there are higher numbers of female veterinarians and animal scientists in Brazil [45].

The variation observed in the answers related to alternatives to conventional meat and Brazilian investment in cell-based meat across respondents with a different frequency of meat consumption (*p* = 0.021 and *p* < 0.001, respectively) seems coherent since vegetarians and vegans are likely to explore more alternatives for animal proteins than meat consumers; therefore, they tend to associate meat with meat analogues such as soy [46,47]. Although this group might have lower interest in consuming cell-based meat [21,46,47,48], they show more support for it [9], probably due to ethical issues [48].

Subsequently, LCA of the first group of questions produced LC 1 (79.8%, %; 217/272), with a tendency to support conventional meat, and LC 2, with a trend to support cell-based meat (20.2%; 55/272). The heterogeneity expressed by both LCs about problems in the production of cell-based meat is probably associated with unfamiliarity with the subject, as seen per the high percentage of “do not know” answers in relation to the problems of cell-based meat. Likewise, other studies with consumers reported that most people did not know [21] or did not have enough knowledge [33] on cell-based meat to criticise it. It is relevant to notice that responses from LC 1 also described problems in the conventional production of meat, indicating that in general, people are aware of the negative impacts of conventional meat [44,49]. On the other hand, LC 2 positive responses to the benefits of cell-based meat may be related to the direct and intrinsic implication of this new method of meat production, which reduces the number of slaughtered animals [50]. Additionally, the greater chances of women belonging to LC 2 may be related to the sensibility of women in relation to animal welfare [51]. Similarly, the lower chances of meat-eaters being in LC 2 may also be related to the lack of knowledge regarding meat substitutes, which might be related to a belief in other solutions such as reducing meat consumption or eating organic foods [16].

LCA for the second group of questions provided a similar pattern for proconventional meat respondents (55.5%; 151/272) in LC 1 and for respondents supporting cell-based meat (44.5%; 121/272) in LC 2. Both classes recognised the benefits of conventional meat to human health, which is likely related to the nutritional value of meat [52,53]. Similarly, they recognised the efficiency of the conventional production of meat. In this case, the perception may be related to the role of the professions surveyed in animal production, the importance of livestock to the Brazilian economy [54] and the significant investments in research for the efficiency of the conventional systems of meat production over the last decades.

Respondents classified as LC 1 showed lower perception in relation to negative environmental and animal welfare impacts of conventional production, despite broad consensus on these issues in the literature [55,56]. Although awareness of the impacts of meat consumption and production to the environment and animal welfare have been raised [49], there remain conflicts of ideas, which may also relate to incoherence between consciousness and actual changes in behaviour [57], such as reducing meat consumption to help the environment. Specifically, in relation to animal welfare issues, responses tend to align with the concept of the “meat paradox” [17,44], in which people eat meat but do not want to harm animals. On the other hand, responses in LC 2 tend to consider cell-based meat as an alternative to the environmental and animal welfare impacts, as found in previous studies with consumers [9,17,33]. Moreover, the majority of respondents marked “do not know” for questions regarding health and production efficiency in relation to cell-based meat. These results were expected, given the initial stage, essentially experimental, of cell-based meat production and the challenges of escalating it [6].

Regarding the demographic characteristics of respondents grouped in each LC, the same pattern as with the first LCA appeared, with women, veterinarians, vegetarians and vegans more likely to belong to LC 2; therefore, they have more positive perceptions related to cell-based meat regarding the environment and animal welfare. According to Ruby and Heine [51], women are more engaged in animal protection and are more concerned with animal welfare and environmental issues, which make them more prone to transition to meatless diets. Likewise, De Backer and Hudders [43] recognised the relationship between animal welfare empathy and meatless diets. Additionally, Sanchez-Sabate and Sabaté [49] noted the willingness to reduce meat for environmental reasons in the Western population. Considering the available knowledge, it seems that respondents in LC 2 may present a higher perception of cell-based meat as an alternative to conventional meat production that is more ethical and less damaging to the environment.

As for the perception of respondents regarding the probable positioning of our target professions, both veterinarians and animal scientists believed that bioprocess engineers will be the most positive professionals for cell-based meat due to their knowledge of the field and the possibility of new jobs. On the other hand, as we hypothesised, veterinarians and animal scientists predicted that their own classes will show resistance to cell-based meat. In their opinions, these professionals may feel threatened by the loss of their work in animal production and tend to assume that they do not have opportunities in the cell-based meat industry. This pattern may reflect the resistance to change as a motivation problem rather than skill [25], as well as the mainstream education and specialisation of those professionals, which focus on well-established knowledge and concepts. This, in turn, may be a major cause for resistance in general when professionals are exposed to new discoveries [58], such as biotechnology. Thus, our results suggest that the use of strategies to stimulate the specialists involved, such as communication of the need for change and group participation in planning the changes [25] and higher education in biotechnology and other subjects related to tissue engineering, may mitigate resistance in both the veterinary and animal science programs.

However, Seth Bannon, one of the investors of the cell-based startup Memphis Meats, reported in the book “Clean Meat: How growing meat without animals will revolutionize dinner and the world” [59] that, “traditionally, we’ve domesticated animals to harvest their cells for food or drink. Now we’re starting to domesticate cells themselves”. Therefore, similar attributions to veterinarians and animal scientists regarding animal production may prospect into cell-based production, such as cellular nutrition and genetics. In addition, Reis et al. (2020) [31] suggested that beyond the demand for biotechnology, there is also a need for business development and management capabilities, knowledge of the food chain, product innovation and networking skills. Therefore, there is a variety of opportunities for the engagement of knowledge stemming from the traditional meat industry.

In the qualitative analysis, the word clouds showed that veterinarians and animal scientists strongly associate cell-based meat with the word “artificial”. Thus, this seems to be a critical point of resistance to this innovation. Mitigation strategies are warranted since the major consequence of negating or slowing-down the development of cell-based meat will likely be the loss of meat market shares and increased dependence on foreign technology. The results seem coherent as the conceptualisation of cell-based meat and unnaturalness is also reported in articles focusing on consumer opinion [17,18,33]. It is noted throughout the literature that the word “artificial” is widely used to refer to cell-based meat, which is a problem since it has a negative connotation [60]. Bryant et al. (2019) [19] suggested that a discussion about the unnaturalness of conventional meat is the best strategy to prevent negative attitudes concerning cell-based meat and its perceived artificiality. Secondly, respondents seemed to worry about the sensory characteristics of cell-based meat, mentioning the words “tasteless” and “flavour”. Likewise, many consumer studies show respondent concerns about the flavour and appearance of cell-based meat, in how well it mimics conventional meat [9,15,16]. Even though many startups have been working on improving the technology to reach best possible consumer acceptance, there are important challenges to be overcome, such as advancements in final product structure to provide different textures to cell-based meat [6]; in this sense, a long way ahead seems needed, considering the diversity of conventional meat products in animal species, breed, as well as the variety of cuts available. Additionally, the word “strange” seems to have a negative connotation, perhaps associated with food neophobia, i.e., the fear of new foods. Thus, Wilks et al. (2019) [61] suggest negative attitudes regarding cell-based meat because it is a novel food and not yet available for human consumption. This may be attenuated by increasing the knowledge of this new production system since we are less likely to fear what we understand [17].

The word clouds presented different characteristics when veterinarian responses were compared to animal scientist ones. First, veterinarians cited a broader range of terms, a fact that may be explained by the higher number of veterinarian respondents. However, more research is needed to better understand the views of each professional group. In addition, the words “animal” and “welfare” appeared more frequently in the veterinarian responses than in those of the animal scientists. This may be related to higher cognitive dissonance, due to the obligatory and perhaps more exclusive animal production contents studied during that course, as similarly reported in consumer studies [16,44].

Finally, this study is highly original in that it presents the first assessment of what those directly involved in the meat production chain think about the new alternatives to conventional meat. However, it also presents important limitations, which warrant further studies. First, our objective is clearly limited, and other relevant aspects of the disruptive innovation of cell-based meat remain worth studying, such as the professional perceptions regarding environmental impacts, public health consequences, and the balance of gains and losses in terms of local jobs and economy, depending on whether a country decides to invest in this new production chain or remain resistant to it. Second, our sample size may not be representative of each professional background studied or of all regions in Brazil, which are diverse in geographical, sociological, educational, cultural and other important aspects. Thus, research that includes respondents from other regions of the country, as well as respondents from other countries, is likely to enrich knowledge by presenting new perceptions and allowing additional conclusions. In addition, the number of veterinarians we were able to engage in our survey was more than three times higher than that of animal scientists, which may have influenced our results. For example, both the number of terms in the word clouds and the number of central ideas in respondent justifications presented more variety for the veterinarian responses, which has likely been affected by the fact that there were more respondents in this category. As an additional level of complexity, respondents were mainly from the South of Brazil, where there are less animal science courses (18/129) than in other regions of the country, according to the National Register of Higher Education Courses and Institutions [57]. This shows an interaction between the differences amongst Brazilian regions and the representation of professional backgrounds. An additional limitation of our results is that they represent respondents of a predominantly young age group; the effects of such a characteristic on our data set deserve further studies, as it may significantly influence responses. Therefore, our conclusions require caution regarding their generalisation, and knowledge in this field is dependent on additional studies.

## 5. Conclusions

This is the first analysis of the perception of veterinarians and animal scientists regarding cell-based meat, approaching a potential bottleneck to the development of animal food production systems which may alleviate most of the animal suffering that exists in intensive industrial animal production chains. Our findings reveal the main reasons for the resistance of veterinarians and animal scientists to cell-based meat in Brazil: lack of knowledge and the perception of a connection between cell-based meat and artificiality. It seems fundamental to generate strategies of motivation and higher education for these professionals in the required capabilities of the cell-based meat chain. Additionally, increasing specific knowledge may foster their engagement in the disruptive innovation represented by cell-based meat, mitigating both the resistance and its negative impacts for the professionals and the animals.

## Figures and Tables

**Figure 1 animals-10-01678-f001:**
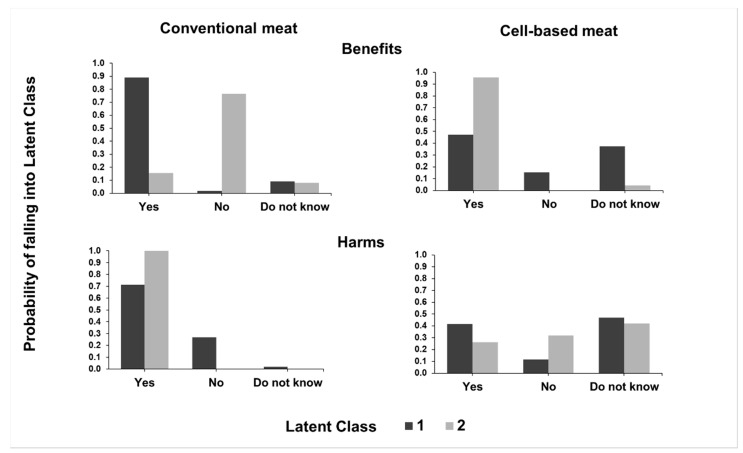
Benefits and harms of conventional and cell-based meat in each latent class according to 209 Brazilian veterinarians and 63 animal scientists responding to an online questionnaire. Latent Class 1 (217 respondents) presents a more proconventional perception, and LC 2 (55 respondents), a more pro-cell-based meat perception.

**Figure 2 animals-10-01678-f002:**
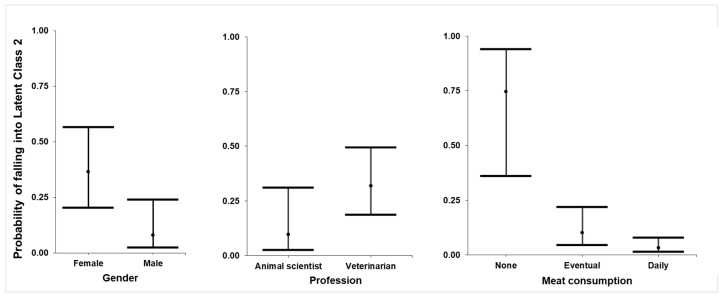
Adjusted respondent probability of falling into Latent Class 2 (pro-cell-based meat, 55 respondents) for the group of questions related to the benefits and harms of conventional and cell-based meat production, according to the responses to an online questionnaire by 209 Brazilian veterinarians and 63 animal scientists.

**Figure 3 animals-10-01678-f003:**
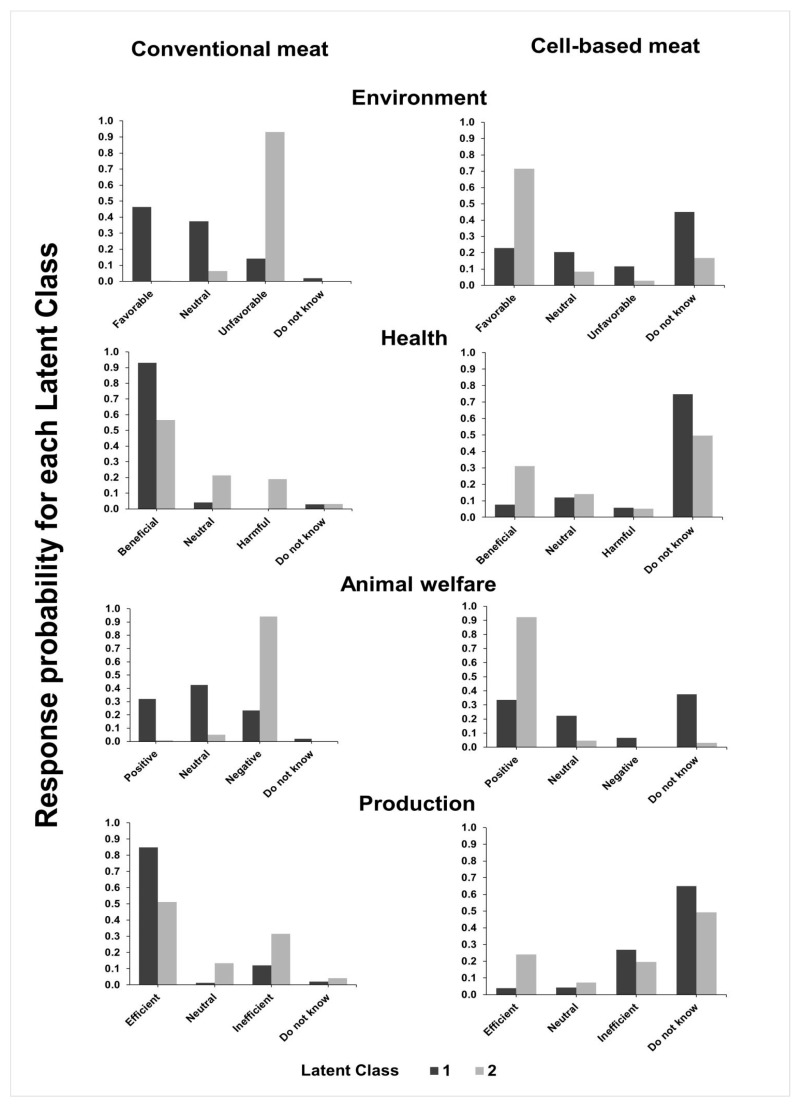
Perceptions of issues related to the environment, health, animal welfare and efficiency of production of conventional and cell-based meat in each latent class (LC) by 209 Brazilian veterinarians and 63 animal scientists according to an online questionnaire. LC 1 (151 respondents) presents a more proconventional perception and LC 2 (121 respondents) a more pro-cell-based perception.

**Figure 4 animals-10-01678-f004:**
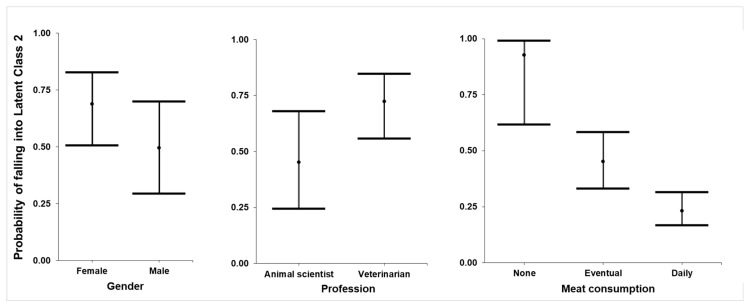
Adjusted respondent probability of falling into Latent Class 2 (pro-cell-based meat, 121 respondents) for the group of questions related to the environment, health, animal welfare and efficiency of production of conventional and cell-based meat, according to the responses to an online questionnaire by 209 Brazilian veterinarians and 63 animal scientists.

**Figure 5 animals-10-01678-f005:**
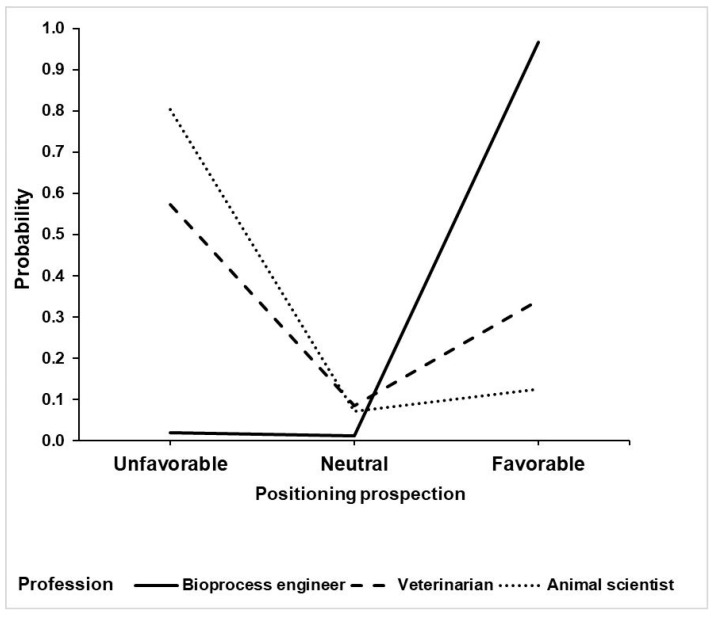
Adjusted logit model for the perception of the probable position that veterinarians, animal scientists and bioprocess engineers will assume regarding the production of cell-based meat, according to an online questionnaire by 209 Brazilian veterinarians and 63 animal scientists.

**Figure 6 animals-10-01678-f006:**
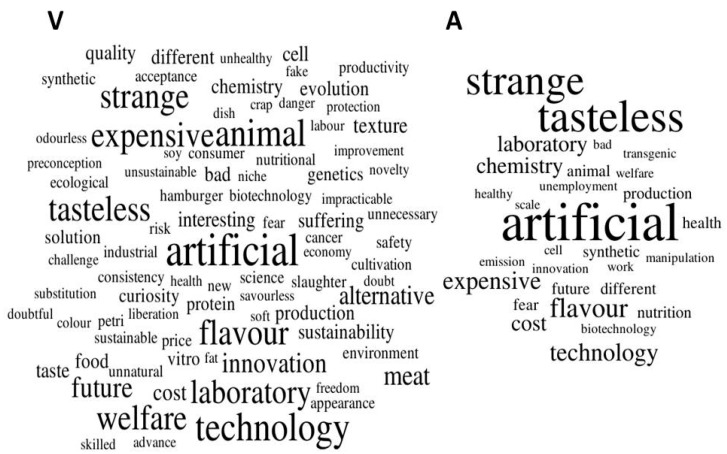
Word clouds composed by words which appeared two or more times in answer to the question “*What comes to your mind when you think of cell-based meat?*”, according to 209 Brazilian veterinarians (V) and 63 animal scientists (A) in a questionnaire.

**Table 1 animals-10-01678-t001:** Results of the logistic regression model for the chances produced by latent class (LC) analysis for the group of questions related to benefits and harms of conventional and cell-based meat, according to the responses to an online questionnaire by 209 Brazilian veterinarians and 63 animal scientists. LC 2 presents a more pro-cell-based perception.

Demographic Data	Contrast	Odds Ratio (LC 2/LC1)	Confidence Interval (95%)
Gender	Men as compared to women	0.131	0.036–0.384
Profession	Veterinarians as compared to animal scientists	5.224	1.483–25.231
Meat consumption	Partial as compared to vegetarians and vegans	0.041	0.008–0.154
Meat consumption	Daily as compared to vegetarians and vegans	0.012	0.002–0.047

**Table 2 animals-10-01678-t002:** Results of the logistic regression model on the chances produced by latent class (LC) analysis for the group of questions related to the environment, health, animal welfare and efficiency of production of conventional and cell-based meat, according to the responses to an online questionnaire by 209 Brazilian veterinarians and 63 animal scientists; Latent Class 2 presents a more pro-cell-based conviction.

Demographic Data	Contrast	Odds Ratio (LC2/LC1)	Confidence Interval (95%)
Gender	Men as compared to women	0.426	0.231–0.754
Profession	Veterinarians as compared to animal scientists	3.291	1.637–6.846
Meat consumption	Partial as compared to vegetarians and vegans	0.069	0.008–0.382
Meat consumption	Daily as compared to vegetarians and vegans	0.024	0.003–0.126

**Table 3 animals-10-01678-t003:** Discourse of the Collective Subject results for answers on the perception of 209 Brazilian veterinarians (V) and 63 animal scientists (A) regarding probable positioning of veterinarians, animal scientists and bioprocess engineers (B) in relation to cell-based meat, according to an online questionnaire.

Respondent/ Question Target	Respondent Perception of Probable Positioning	Central Idea	Examples of Excerpts in the Original Text of Respondents	Central Ideas Frequency
V/V	Unfavourable (*n* = 65)	field of work	“Meat production systems are areas of practice of the veterinarian”	25
		job losses	“They will think this is a threat to the jobs for that class”	28
		tradition	“Many veterinarians are not opened to novelties and are reluctant to accept …”	16
		other ideas		21
	Neutral (*n* = 23)	field of work	“It depends on the field of work”	4
		more study	“It depends on more studies”	3
	Favourable (*n* = 22)	animal welfare	“More and more professionals in the area are concerned with animal welfare...”	15
		job offers	“… animal protein production is broad and not restricted only to the farm and production …”	5
		health and safety	“… rigorous quality control which increases the quantity and reduces the risks of the product”	4
		other ideas		4
V/A	Unfavourable (*n* = 65)	field of work	“They are the professionals that act mainly in animal production”	35
		job losses	“Reduction of demand for services”	25
		tradition	“Old concepts”	7
		other ideas		9
	Neutral (*n* = 12)	no changes	“It will not affect the profession”	2
		field of work	“… many professionals will be conflicted because they depend on animal production”	2
		other ideas		6
	Favourable (*n* = 8)	new area	“Opportunity for a new area of activity”	3
		animal welfare	“Aims at animal welfare”	2
		environment	“Because it is important to seek alternatives to reduce the environmental impact of agriculture”	2
		other ideas		1
V/B	Unfavourable (*n* = 5)	unfamiliarity	“It is not meat”	1
		field of work	“… these professionals also encompass the improvements to conventional meat production systems”	1
	Neutral (*n* = 1)	more studies	“We must wait for the evaluation of results”	1
	Favourable (*n* = 79)	field of work	“… will obtain the greatest benefit because this field is essential …”	40
		opportunities	“The technology tends to be more interesting for this group”	22
		job offers	“It will expand their labour market”	19
		other ideas		2
A/V	Unfavourable (*n* = 12)	field of work	“The animal production is part of the attributions of these professionals”	4
		job losses	“It will directly reach the jobs of these professionals”	3
		other ideas		5
	Neutral (*n* = 4)	field of work	“Those who work in production will be unfavourable. Who works with pets will be favourable”	2
		other ideas		2
	Favourable (*n* = 6)	animal welfare	“The slaughter of animals will decrease”	3
		other ideas		3
A/A	Unfavourable (*n* = 27)	field of work	“They will feel their profession is under threat”	12
		job losses	“Because they cannot work in the meat labs”	6
		quality of cell-based meat	“We do not have studies guaranteeing the quality of this product”	4
		benefits of conventional meat	“Because they know the importance of animal production for the functioning of societies and nature”	3
		other ideas		1
	Neutral (*n* = 5)	emergence of market	“I believe that cell-based meat will serve another group of consumers, which has been increasing considerably, not interfering with consumers of conventional meat”	5
		animal welfare	“It stimulates the consumption of meat by niches concerned with animal welfare”	2
	Favourable (*n* = 3)	technology	“After some time, the production technologies will be consolidated”	2
		job offers	“Another job opportunity”	1
A/B	Unfavourable (*n* = 3)	negative image of conventional meat	“Propagation of a product in which people will make an even more negative image of conventional meat …”	1
		competition	“… Many of these professionals are in the same job market as veterinarians and animal scientists”	1
	Neutral (*n* = 1)	field of work	“Another area of work”	1
	Favourable (*n* = 22)	field of work	“Because it will expand its area of operation”	15
		job offer	“Possibility of working in this area and new job offers”	7
		other ideas		2

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
