# Peer review of "Critical Perspective of Animal Production Specialists on Cell-Based Meat in Brazil: From Bottleneck to Best Scenarios"

_animals, 2020, doi:10.3390/ani10091678_

Round 1
Reviewer 1 Report
Overall, the study is interesting and well constructed. Here some minor considerations required.
Line 160: You say that “only years were considered” but you mean the years from graduation or the age of participants?
Line 180: I would suggest using “ ” around the scale of assessment.
Line 233-234: The sentence doesn’t sound well, please consider rephrasing.
Material and methods:
M&m were clearly stated. Anyway, reading the following section of the manuscript I have seen some discussion about bioprocess engineers. I have not fully understood how many were interviewed in this study since they were mentioned in the “2.1 Development of the research instrument”, but in the “2.2 Survey sample” they were not mentioned (lines 145-16). Please clarify.
Discussion section:
The discussion section is, in my point of view, interesting and well-constructed. Despite this, I would suggest to add a small paragraph about the limitation of the study. For example, the number of professionals involved. As stated in the m&m, veterinary was the most represented category, and sometimes the sample size would influence the results.
Moreover, have you considered the gender of professionals in each category (and not overall?). For example, if animal scientists were mostly men (as occurs in some countries worldwide), should also that (together with the different educational backgrounds, which was correctly stated by the Authors) that could have contributed to the outcomes?
Author Response
Review Report Form
Open Review
(x) I would not like to sign my review report
( ) I would like to sign my review report
English language and style
( ) Extensive editing of English language and style required
( ) Moderate English changes required
(x) English language and style are fine/minor spell check required
( ) I don't feel qualified to judge about the English language and style
Yes Can be improved Must be improved Not applicable
Does the introduction provide sufficient background and include all relevant references?
(x) ( ) ( ) ( )
Is the research design appropriate?
(x) ( ) ( ) ( )
Are the methods adequately described?
( ) (x) ( ) ( )
Are the results clearly presented?
(x) ( ) ( ) ( )
Are the conclusions supported by the results?
(x) ( ) ( ) ( )
Comments and Suggestions for Authors
Overall, the study is interesting and well constructed. Here some minor considerations required.
Line 160: You say that “only years were considered” but you mean the years from graduation or the age of participants?
R: We emphasized here that regarding all variables analysed, we only considered years since graduation for further analyses due to their strong correlation (line 183).
Line 180: I would suggest using “ ” around the scale of assessment.
R: Thank you for your suggestion. We inserted the quotation marks in line 203-204.
Line 233-234: The sentence doesn’t sound well, please consider rephrasing.
R: We improved the sentence to clarify it (line 257-259).
Material and methods:
M&m were clearly stated. Anyway, reading the following section of the manuscript I have seen some discussion about bioprocess engineers. I have not fully understood how many were interviewed in this study since they were mentioned in the “2.1 Development of the research instrument”, but in the “2.2 Survey sample” they were not mentioned (lines 145-16). Please clarify.
R: In this paper, our aim was to study veterinarian and animal scientists’ opinion regarding cell-based meat, as we mentioned in lines 124-126. In 2.1 we addressed the questions considered for analysis in the present paper. In lines 144-146 we featured the questions in respect of respondent perception regarding the probable positioning of veterinarians, animal scientists and bioprocess engineers in relation to cell-based meat. Therefore, in 2.2 we emphasized that only veterinarians and animal scientists were considered for the survey sample (lines 165-166). Thus, the discussion involving bioprocess engineers is based on the veterinarian and animal scientists’ perceptions on how bioprocess engineers will react to cell-based meat.
Discussion section:
The discussion section is, in my point of view, interesting and well-constructed. Despite this, I would suggest to add a small paragraph about the limitation of the study. For example, the number of professionals involved. As stated in the m&m, veterinary was the most represented category, and sometimes the sample size would influence the results.
R: Thank you for your suggestion. We added a paragraph in the discussion (lines 528-551) presenting the limitations of the study.
Moreover, have you considered the gender of professionals in each category (and not overall?). For example, if animal scientists were mostly men (as occurs in some countries worldwide), should also that (together with the different educational backgrounds, which was correctly stated by the Authors) that could have contributed to the outcomes?
R: Thank you for your comment. We detailed the number of women and men for each professional in item 2.2 (lines 169-172). Thus, we stated the majority of women on both veterinarian and animal science’s programs in Brazil (lines 418-419) and that may be related to the higher number of female respondents.
Reviewer 2 Report
Review, paper no. animals-912385 “„Critical perspective of animal production specialists on cell-based meat: from bottleneck to best scen arios”. Generally, the article has some interesting findings which could be worth publication. Besides winning the favour of animal rights activists for its humane production of meat, in vitro meat production system also circumvents many of the issues associated with conventional meat production systems, like excessively slaughter, foodborne illnesses, antibiotic-resistant pathogen strains, and emissions of GG that contribute to global warming. There is no discussion in the manuscript on the impact of cell-based meat on the economic condition of meat producers (farmers). This is an important public (social) aspect. The title of the work and aim should be noted that the study focused on Brazil.
Generalny, consumer acceptance will be strongly influenced by many factors and consumers seem to dislike unnatural food. This aspect is not discussed at work. A survey template should be addend (Supplementary Material). Some minor comments provided below.
Specific comments
L. 28-29. „absolute farm animal welfare development” - This sentence should be rewritten.
L.23-24; L. 39. What do you mean „negative impacts for the animals and the environment”. Rewrite this sentence.
L. 51. Remove „an individual”.
L. 64. Add a discussion of the negative aspects (health, producer economics).
L. 101 -103. It should be noted that the study focused on Brazil.
L. 148. The survey involved „young people" - should be subjected to an objective assessment in the discussion and summary.
L. 194. There are no citations in the literature packet R.
The statistical analysis seems correct.
L. 246. Date are unnecessary.
L. 264. Date are unnecessary.
L. 276. Date are unnecessary.
L. 297. Date are unnecessary.
L. 304. Date are unnecessary.
L. 317. Date are unnecessary.
L. 337. Date are unnecessary.
L. 350. Date are unnecessary.
Figure 6 is correct. It's a graphic abstrakt.
L. 371. Date are unnecessary.
L. 392. Sentence not clear. What are the programs, what are they about, are they related to the topic.
L.409. Define if appropriate (LCA, LC).
L. 506. First analysis in Brazil.
L. 496. In terms of technical issues, research is still required to optimize cell culture methodology. Please add a discussion. It is also almost impossible (or very complicated) to reproduce the diversity of meats derived from various species, breeds and cuts.
L. 509-511. Authors should better summarize the results and the significance of the study in this section.
Author Response
Review Report Form
Open Review
(x) I would not like to sign my review report
( ) I would like to sign my review report
English language and style
(x) Extensive editing of English language and style required
( ) Moderate English changes required
( ) English language and style are fine/minor spell check required
( ) I don't feel qualified to judge about the English language and style
Yes Can be improved Must be improved Not applicable
Does the introduction provide sufficient background and include all relevant references?
( ) (x) ( ) ( )
Is the research design appropriate?
(x) ( ) ( ) ( )
Are the methods adequately described?
(x) ( ) ( ) ( )
Are the results clearly presented?
(x) ( ) ( ) ( )
Are the conclusions supported by the results?
( ) (x) ( ) ( )
Comments and Suggestions for Authors
Review, paper no. animals-912385 “„Critical perspective of animal production specialists on cell-based meat: from bottleneck to best scen arios”. Generally, the article has some interesting findings which could be worth publication. Besides winning the favour of animal rights activists for its humane production of meat, in vitro meat production system also circumvents many of the issues associated with conventional meat production systems, like excessively slaughter, foodborne illnesses, antibiotic-resistant pathogen strains, and emissions of GG that contribute to global warming. There is no discussion in the manuscript on the impact of cell-based meat on the economic condition of meat producers (farmers). This is an important public (social) aspect. The title of the work and aim should be noted that the study focused on Brazil.
R: Thank you for your comments. We focused our discussion on the aspects related to the professional implications with the introduction of cell-based meat in the meat chain. Thus, we did not discuss possible economic and social aspects due to the limitations of our objectives, as we feel these are complex issues that deserve in depth research. However, we agree that this aspect is very important and should be analysed in future studies, and we have mentioned this in the approach of the limitations of our study (lines 528-551). In regards to the title and the aim of our paper, we appreciate your attention to this aspect and we have adopted your suggestions.
Generalny, consumer acceptance will be strongly influenced by many factors and consumers seem to dislike unnatural food. This aspect is not discussed at work. A survey template should be addend (Supplementary Material). Some minor comments provided below.
R: Thank you for your comments. We are aware of the importance of the unnaturalness aspect of cell-based meat. Thus, we briefly mentioned the influence of unnaturalness in line 79, followed by a discussion of unnaturalness associated with artificiality and a possible strategy to reduce its negative impacts on lines 505-507. As for these aspects in terms of Brazilian consumers, we can actually go a bit further than making general inferences, as local data is available in Valente et al, 2019, a reference we have cited in 80, 117, 425. Finally, in relation to the supplementary material, our data was collected in Portuguese.
Specific comments
- 28-29. „absolute farm animal welfare development” - This sentence should be rewritten.
R: Thank you for your suggestion. We improved the sentence in line 29-30.
L.23-24; L. 39. What do you mean „negative impacts for the animals and the environment”. Rewrite this sentence.
R: Thank you for your comment. These statements include major assumptions for the very motivation to conduct the survey and we believe they are essential to our closing, as they contextualize the relevance of our study within the bigger scenario. In attention to your comment, we substituted the word “impacts” for the word “consequences” (line 40); we hope this may in part address your concern while simultaneously retaining what is important for the authors.
- 51. Remove „an individual”.
R: Thank you for your suggestion. We removed the words as suggested and rearranged the phrase accordingly (line 51).
- 64. Add a discussion of the negative aspects (health, producer economics).
R: Thank you for your suggestion. We improved the discussion of the negative aspects as suggested (lines 64-68).
- 101 -103. It should be noted that the study focused on Brazil.
R: Thank you for your attention to this detail. We inserted the focus on Brazil in the title and the objective (lines 124-126).
- 148. The survey involved „young people" - should be subjected to an objective assessment in the discussion and summary.
R: Thank you for this comment. We have accordingly considered that in the discussion section (lines 416-418). In addition, we have inserted a discussion regarding respondent predominant age group again in the paragraph on study limitations (lines 528-551). We did not insert it in the summary as it would unbalance it, in the sense that other variables at the same level of discussion would also be required; however, there is not room for these to be included in the abstract.
- 194. There are no citations in the literature packet R.
R: We added the references for the packet R and poLCA in the list (references 37-38).
The statistical analysis seems correct.
R: Thank you.
- 246. Date are unnecessary.
- 264. Date are unnecessary.
- 276. Date are unnecessary.
- 297. Date are unnecessary.
- 304. Date are unnecessary.
- 317. Date are unnecessary.
- 337. Date are unnecessary.
- 350. Date are unnecessary.
- 371. Date are unnecessary.
R: Thank you for your attention to details. We removed the dates mentioned in description of tables and figures.
Figure 6 is correct. It's a graphic abstrakt.
R: Thank you.
- 392. Sentence not clear. What are the programs, what are they about, are they related to the topic.
R: Thank you for your comment. We added information as requested (line 403-410).
L.409. Define if appropriate (LCA, LC).
R: Thank you, we have searched the full paper for adequacy of acronym presentation.
- 506. First analysis in Brazil.
R: In the conclusion, we emphasized the originality of the paper, considering there are no previous studies regarding professional perceptions in relation to cell-based meat from any country, to the best of our knowledge.
- 496. In terms of technical issues, research is still required to optimize cell culture methodology. Please add a discussion. It is also almost impossible (or very complicated) to reproduce the diversity of meats derived from various species, breeds and cuts.
R: Thank you for your suggestion. We improved the discussion regarding cell-based meat technology challenges to overcome (line 510-514).
- 509-511. Authors should better summarize the results and the significance of the study in this section.
R: Thank you for your suggestion. We have summarized the results following our goal with this paper. Thus, first, we explored and compared the perception of Brazilian veterinarians and animal scientists regarding cell-based meat, who showed resistance; then, we presented these points of resistance, related with the lack of knowledge and the association with artificiality. Last, we suggested strategies for mitigating the resistance.

Reviewer 3 Report
Dear authors,
The perception of cell-based meat by professionals in Brazil is described in an interesting way, which is of great guiding significance to the industry and scientists. I reckon that a big effort was done to build this article. Congratulations to you. However, I think there are a few minor problems that need to be revised here.
- The survey was conducted among Brazilian specialists. Therefore, the title of the manuscript should include such a limitation, e.g. from a Brazilian point of view.
- In the introduction, it would be useful to give information in a few sentences on the scale of production of cell-based meat in Brazil or worldwide for consumption or industrial purposes.
- In general, I think the article is too long - please shorten it. Perhaps it could be shortened, for example, by creating an appendix with less relevant results.
- Table 3 is too long, and requires better formatting. Can you move it to the appendix ?
- L. 392. What does that mean? What programs, at what level of education. That needs to be clarified.
- Discussion. Please remove the gaps between the individual paragraphs.
- There are no research limitations in the results or discussion. It’s need to introduce.
Author Response
Review Report Form
Open Review
(x) I would not like to sign my review report
( ) I would like to sign my review report
English language and style
( ) Extensive editing of English language and style required
( ) Moderate English changes required
(x) English language and style are fine/minor spell check required
( ) I don't feel qualified to judge about the English language and style
Yes Can be improved Must be improved Not applicable
Does the introduction provide sufficient background and include all relevant references?
( ) (x) ( ) ( )
Is the research design appropriate?
(x) ( ) ( ) ( )
Are the methods adequately described?
(x) ( ) ( ) ( )
Are the results clearly presented?
( ) (x) ( ) ( )
Are the conclusions supported by the results?
(x) ( ) ( ) ( )
Comments and Suggestions for Authors
Dear authors,
The perception of cell-based meat by professionals in Brazil is described in an interesting way, which is of great guiding significance to the industry and scientists. I reckon that a big effort was done to build this article. Congratulations to you. However, I think there are a few minor problems that need to be revised here.
R: Thank you for your comments and support of this paper. We have given full attention to all suggestion, which we believe have greatly improved our paper.
The survey was conducted among Brazilian specialists. Therefore, the title of the manuscript should include such a limitation, e.g. from a Brazilian point of view.
R: Thank you for your suggestion. We modified the title and aim to emphasize the study was done in Brazil.
In the introduction, it would be useful to give information in a few sentences on the scale of production of cell-based meat in Brazil or worldwide for consumption or industrial purposes.
R: Thank you for your suggestion. We improved the introduction with the description of Brazil in regards to cell-based meat (lines 107-122).
In general, I think the article is too long - please shorten it. Perhaps it could be shortened, for example, by creating an appendix with less relevant results.
Table 3 is too long, and requires better formatting. Can you move it to the appendix ?
R: We agree that Table 3 is rather long. However, we prefer maintaining it in the main text as it provides important information for the understanding of the paper and it is easier for the reader if the table is promptly available. In addition, other reviewers seem to count on it in the final version, as they have not recommended its transfer to the appendix. We have improved its formatting by merging the two columns to the right into a single one and carefully changing column widths.
- 392. What does that mean? What programs, at what level of education. That needs to be clarified.
R: Thank you for your comment. We added information as requested (line 403-410).
Discussion. Please remove the gaps between the individual paragraphs.
R: Thank you for your attention to details. We correct the paragraphs’ gaps.
There are no research limitations in the results or discussion. It’s need to introduce.
R: Thank you for your suggestion. We added a paragraph in the discussion (lines 528-551) presenting the limitations of the study.
